# PoLi-RL: A Point-to-List Reinforcement Learning Framework for Conditional Semantic Textual Similarity

**Zixin Song**[1*], **Bowen Zhang**[1*], **Qian-Wen Zhang**[2] , **Di Yin**[2] , **Xing Sun**[2] , **Chunping Li**[1] [†]
[1]Tsinghua University, [2]Tencent Youtu Lab
{songzx24,zbw23}@mails.tsinghua.edu.cn,cli@tsinghua.edu.cn
{cowenzhang,endymecyyin,winfredsun}@tencent.com

## Abstract

Conditional Semantic Textual Similarity (C-STS) measures the semantic proximity between text segments under a specific condition, thereby overcoming the ambiguity inherent in traditional STS. However, existing methods are largely confined to discriminative models, failing to fully leverage recent breakthroughs in the NLP community involving Large Language Models (LLMs) and Reinforcement Learning (RL). RL is a particularly well-suited paradigm for this task, as it can directly optimize the non-differentiable Spearman ranking metric and guide the reasoning process required by C-STS. Nevertheless, we find that naively applying listwise RL fails to produce meaningful improvements, as the model struggles with complex, coarse-grained reward signals, leading to optimization difficulties. To address this challenge, we introduce PoLi-RL, a novel Point-to-List Reinforcement Learning framework. PoLi-RL employs a two-stage curriculum: it first trains the model with a simple pointwise reward to establish fundamental scoring capabilities, then transitions to a hybrid reward that combines pointwise, pairwise, and listwise objectives to refine the model's ability to discern subtle semantic distinctions. Crucially, we propose an innovative Parallel Slice Ranking Reward (PSRR) mechanism that computes ranking rewards in parallel slices, where each slice consists of completions with the same index from different samples. This provides a precise, differentiated learning signal for each individual completion, enabling granular credit assignment and effective optimization. On the official C-STS benchmark, PoLi-RL achieves a Spearman correlation coefficient of 48.18, establishing a new SOTA for the cross-encoder architecture. Furthermore, PoLi-RL also maintains SOTA status on the re-annotated C-STS dataset, confirming its robust generalization capabilities. As the first work to successfully apply RL to C-STS, our study introduces a powerful paradigm for aligning LLMs for complex, ranking-based conditional judgment tasks. Our code and checkpoints are available at https://github.com/ZBWpro/PoLi-RL.

## 1 Introduction

As a core research area in Computational Linguistics, Semantic Textual Similarity (STS) finds extensive applications across diverse scenarios, including topic modeling, dialogue systems, text summarization, and agent memory (Tang et al., 2025; Zhang et al., 2024b; 2025a). However, traditional STS tasks exhibit inherent ambiguity because similarity definitions are often susceptible to subjective bias. To address this limitation, the Conditional Semantic Textual Similarity (C-STS) task was developed (Deshpande et al., 2023). By incorporating an explicit natural language condition, C-STS enables more precise and objective similarity judgments, yet simultaneously imposes higher demands on a model's reasoning capabilities. For instance, consider the following two text fragments: "A player is shooting from beyond the three-point line" and "A player is taking a free throw". Under

---

*Equal contribution. Work done during the internship at Tencent Youtu Lab.
†Corresponding author.

the condition "The activity of the player", their similarity is high. However, under the condition "The player's distance from the basket", their similarity is low.

Research on this nascent task has yielded three primary paradigms: Bi-encoder (Liu et al., 2025), Tri-encoder (Lin et al., 2024), and Cross-encoder (Li et al., 2024). The Cross-encoder architecture, which processes the text pair and the guiding condition simultaneously, is inherently compatible with modern generative pre-trained models. Despite this, the integration of C-STS with LLMs remains in its early stages. Current LLM applications are limited to two main approaches: direct inference via few-shot prompting, where even state-of-the-art models struggle to achieve satisfactory results (Deshpande et al., 2023); and utilizing them as feature extractors for generating text embeddings (Yamada & Zhang, 2025), which is an extension of the discriminative paradigm. To the best of our knowledge, no prior work has implemented an end-to-end LLM-based cross-encoder for the C-STS task, nor has any integrated it with advanced training techniques like reinforcement learning (RL), leaving a significant research gap.

This paper aims to fill this gap. We argue that incorporating RL into an LLM-based cross-encoder paradigm is a natural fit. This is reflected in two key aspects: First, C-STS requires sophisticated, scenario-based reasoning. For example, in the basketball case described earlier, to correctly assess similarity under the "distance" condition, the model must transcend surface-level semantics to identify the underlying spatial relationship between 'beyond the three-point line' and 'at the free throw line', a process demanding strong abstraction and inference. RL, through its explicit reward signals, can more effectively guide the reasoning process of LLMs (Guo et al., 2025). Second, from an optimization standpoint, RL aligns closely with the task's evaluation criteria. The Spearman correlation coefficient (Zar, 2005), a core evaluation metric of C-STS, is a non-differentiable measure of ranking quality. Traditional SFT methods can only indirectly and approximately optimize this objective through loss functions like Mean Squared Error (MSE) (Zhang & Li, 2024b). In contrast, RL allows for the direct optimization of ranking-based reward functions designed to correlate strongly with the final Spearman metric, aligning the training objective more closely with the evaluation criteria.

However, a naive application of RL to this task presents significant challenges. As illustrated in Figure 1, our preliminary experiments indicate that directly applying a listwise ranking reward (e.g., Spearman's correlation coefficient) across an entire batch of completions does not show any advantages compared to the few-shot method. This approach suffers from two fundamental problems. First, the ranking objective is too complex for a model that has not yet grasped the task's fundamental scoring rules, often leading to training collapse. Second, a single reward computed across the entire batch is too coarse to provide precise credit assignment, as a few poor completions can unfairly penalize other good ones.

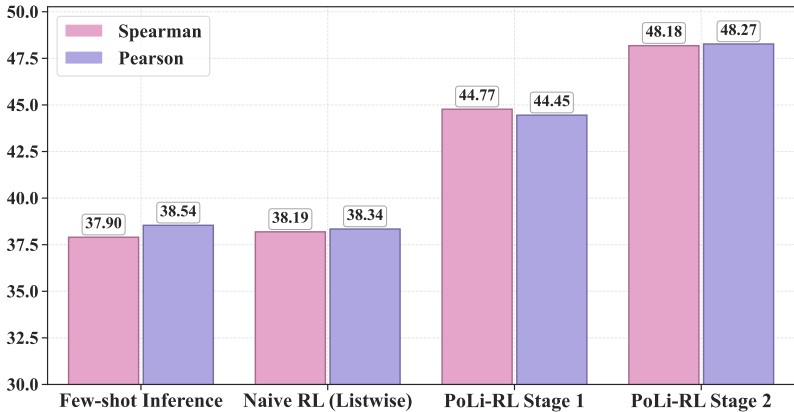

Figure 1: Performance comparison of different strategies on the C-STS task. Directly applying a listwise ranking reward for RL does not significantly outperform the few-shot baseline. In contrast, both stages of our method (PoLi-RL) achieve substantial improvements, validating its effectiveness.

To address these challenges, we propose **PoLi-RL**, a two-stage **Po**int-to-**Li**st **R**einforcement **L**earning framework. PoLi-RL features a two-stage curriculum to manage the complexity of the learning task. In the first stage, we use a simple pointwise reward to ground the model in the basic

scoring rules of the task. Building on this foundation, the second stage introduces a richer, hybrid reward signal that combines a robust pointwise anchor with more nuanced pairwise and listwise ranking rewards, enabling the model to discern subtle semantic differences.

Furthermore, to resolve the problem of coarse-grained reward signals that arises from ranking all completions in a batch together, we innovatively introduce a Parallel Slice Ranking Reward (PSRR) mechanism, which employs a two-level decomposition. First, for a batch of $N$ input samples, the model generates $G$ completions for each. We then form $G$ "parallel slices", where the $i$-th slice is composed of the $i$-th completion from every sample. Second, and more importantly, within each slice, rather than assigning a single reward, we compute the rank difference for each individual completion against its ideal rank. This two-level decomposition allows each of the $N \times G$ completions to receive a unique and precise reward that reflects its quality, thereby enabling granular credit assignment and more effective optimization.

The primary contributions of this paper are outlined as follows:

- To the best of our knowledge, this is the first work to propose an end-to-end, LLM-based cross-encoder for the C-STS task, and also represents the first application of reinforcement learning in this domain.

- We design and implement PoLi-RL, a novel two-stage training curriculum that overcomes the optimization difficulties of direct rank-based learning by progressing from a basic pointwise objective to a comprehensive hybrid reward.

- We propose the Parallel Slice Ranking Reward (PSRR) mechanism, which delivers precise and differentiated learning signals by computing ranking rewards within independent 'parallel slices.' This mechanism offers a generalizable strategy for other ranking tasks involving multiple candidate generations.

- On the official C-STS benchmark, PoLi-RL achieves a Spearman's correlation coefficient of 48.18, establishing a new SOTA for the cross-encoder architecture and surpassing strong proprietary models, including GPT-4o (44.23) and Deepseek-R1 (42.85). Furthermore, achieving SOTA on the re-annotated dataset and our qualitative analysis jointly confirm the model's superior capability in nuanced conditional reasoning.

## 2 METHODOLOGY

This section details our proposed strategy. We begin by formulating the C-STS task within an end-to-end, LLM-based cross-encoder paradigm in subsection 2.1. Then, in subsection 2.2, we map this task onto the mathematical framework of Reinforcement Learning and specify its optimization objective. Finally, in subsection 2.3, we provide a comprehensive description of our PoLi-RL framework, including its two-stage design and the innovative PSRR mechanism.

### 2.1 PROBLEM FORMULATION

The core objective of C-STS is to learn a scoring function that accurately reflects the semantic similarity between two text segments under a specific condition. Formally, each C-STS data sample is defined as a pair $(x, y)$, where the input $x = (t_1, t_2, c)$ consists of two text segments and a natural language condition, and the label $y \in [1, 5]$ is the human-annotated similarity judgment on the Likert scale (Likert, 1932). Specifically, the label $y$ corresponds to a fine-grained set of semantic criteria. According to the C-STS annotation guidelines (Deshpande et al., 2023), the scores represent: (1) Completely dissimilar; (2) Thematically related but dissimilar; (3) Roughly equivalent, but with some important information differences; (4) Mostly equivalent, differing only in unimportant details; (5) Completely equivalent. This level of granularity demands that the model perform fine-grained reasoning beyond surface-level semantics, posing a significant challenge to its capabilities.

A distinctive feature of the C-STS dataset is its paired structure: samples are organized in adjacent pairs that share the same text segments $(t_1, t_2)$ but feature different conditions and maintain a strict ordinal relationship between their labels, i.e., $y_{\text{high}} \geq y_{\text{low}}$. This structure provides a solid foundation for our pairwise reward design, as detailed in subsection 2.3.

Our task is to train a scoring model $\pi_\theta$, parameterized by $\theta$. For each sample, the model takes a unified prompt $p = [\mathcal{I}, \mathcal{E}, x]$ (detailed in Appendix A.2) as input, consisting of the instruction $\mathcal{I}$, $K$ few-shot demonstrations $\mathcal{E} = \{(x_k, y_k)\}_{k=1}^K$, and the query input $x$, to generate an output sequence $o = \pi_\theta(p)$. From this sequence, we parse the final predicted score, $\tilde{y} = \text{Parse}(o)$. The overall training objective is to maximize the ranking consistency between the set of predicted scores $\{\tilde{y}_i\}_{i=1}^N$ and the ground-truth scores $\{y_i\}_{i=1}^N$ by optimizing the policy $\pi_\theta$, a process primarily quantified by Spearman's correlation coefficient. Since this metric is rank-based and non-differentiable, RL emerges as a more promising optimization paradigm than traditional supervised fine-tuning.

## 2.2 Reinforcement Learning for C-STS

We formulate the C-STS task as a Markov Decision Process (MDP), defined by a tuple $\mathcal{M} = (\mathcal{S}, \mathcal{A}, \mathcal{T}, \mathcal{R}, \gamma)$, where the agent is the LLM policy $\pi_\theta$. The generation process is formulated as a sequential decision-making process, where each step involves generating a single token. A state $s_t \in \mathcal{S}$ at timestep $t$ is the sequence of tokens generated so far, conditioned on the initial prompt, i.e., $s_t = (p, o_{<t})$. An action $a_t \in \mathcal{A}$ is the selection of the next token $o_t$ from the model's vocabulary, governed by the policy $\pi_\theta(a_t|s_t)$, which provides a probability distribution over all possible tokens. The transition function $\mathcal{T}$ is deterministic, where the next state $s_{t+1}$ is formed by appending the selected token $a_t$ to $s_t$. We employ a terminal reward setting, where a reward $\mathcal{R}_T = \mathcal{R}(x, y, o)$ is given only after the entire sequence $o$ has been generated. Finally, the discount factor $\gamma$ is set to 1 to ensure that the terminal reward is backpropagated without decay to all actions that contributed to the final output. Based on this framework, the objective is to find the optimal parameters $\theta^*$ that maximize the expected reward over the data distribution $\mathcal{D}$:

$$\theta^* = \arg\max_\theta \mathbb{E}_{(x,y)\sim\mathcal{D}, o\sim\pi_\theta(p)}[R(x, y, o)] \tag{1}$$

To optimize this objective, we employ Decoupled Clip and Dynamic Sampling Policy Optimization (DAPO) (Yu et al., 2025), an extension of GRPO (Shao et al., 2024) that introduces several key techniques for effective RL. For each sample $x$, the policy model generates a set of $G$ completions $\{o_i\}_{i=1}^G$. A scalar reward $r_i = R(x, y, o_i)$ is computed for each completion. The advantage $\hat{A}_i$ for each completion is then calculated by normalizing its reward against the statistics of the group rewards via Z-score normalization:

$$\hat{A}_i = \frac{r_i - \text{mean}(\{r_i\}_{i=1}^G)}{\text{std}(\{r_i\}_{i=1}^G) + \epsilon} \tag{2}$$

Since we adopt an on-policy training approach, completions are sampled directly from the current policy $\pi_\theta$ being optimized. The objective function for updating the model parameters $\theta$ is:

$$\mathcal{J}_{\text{DAPO}}(\theta) = \mathbb{E}_{(x,y)\sim\mathcal{D}, \{o_i\}_{i=1}^G \sim \pi_\theta(\cdot|p)} \left[ \frac{1}{\sum_{i=1}^G |o_i|} \sum_{i=1}^G \sum_{t=1}^{|o_i|} \left( \frac{\pi_\theta(o_{i,t}|p, o_{i,<t})}{[\pi_\theta(o_{i,t}|p, o_{i,<t})]_{\text{nograd}}} \hat{A}_i \right) \right] \tag{3}$$

where $[\cdot]_{\text{nograd}}$ stops gradient flow through the denominator, ensuring only the numerator is updated.

## 2.3 PoLi-RL: A Two-Stage Reinforcement Learning Framework

To optimize effectively, we propose PoLi-RL, a framework built on a two-stage progressive reward curriculum. This subsection elaborates our pipeline and its reward mechanisms (Figure 2).

**Stage I: Foundational Skill Acquisition.** The goal of Stage I is to ground the model in the fundamental scoring rules of the C-STS task. For each input sample, the LLM policy generates $G$ completions, from which we parse a set of predicted scores $\{\tilde{y}_j\}$. The total reward for Stage I, $R_{S1}$, is a weighted sum of three components:

$$R_{S1} = \lambda_1 R_{\text{pointwise}} + \lambda_2 R_{\text{binary}} + \lambda_3 R_{\text{format}} \tag{4}$$

The Pointwise Accuracy Reward ($R_{\text{pointwise}}$) is the primary component in Stage I. It measures the normalized distance between the predicted score $\tilde{y}_j$ and the ground-truth score $y_j$:

$$R_{\text{pointwise}} = 1 - \frac{|\tilde{y}_j - y_j|}{\max(Y) - \min(Y)} \tag{5}$$

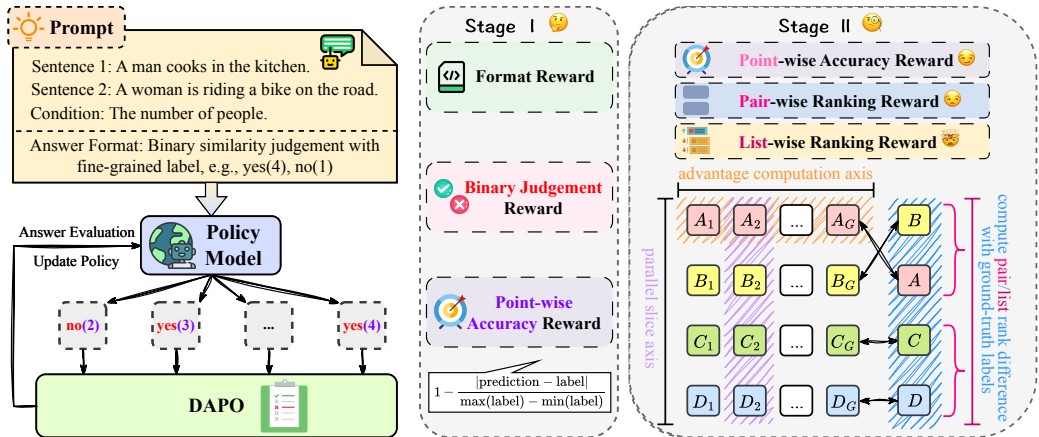

Figure 2: An overview of the PoLi-RL framework. It employs a two-stage curriculum, progressing from Stage I, where the model learns foundational scoring rules, to Stage II, which refines the model's ability to discern fine-grained semantic differences. The core of our method is the PSRR mechanism in Stage II, where pairwise and listwise ranking rewards are computed vertically within slices of the same-indexed completions to provide precise, differentiated learning signals.

where $\max(Y) = 5$ and $\min(Y) = 1$ are the bounds of the label space.

To mitigate the model's tendency to converge to safe, intermediate scores, we introduce a Binary Judgment Reward ($R_{\text{binary}}$). According to the C-STS guideline that scores $\geq 3$ indicate similarity, while scores $\leq 2$ indicate dissimilarity, this reward encourages the model to first master this basic binary classification:

$$R_{\text{binary}} = \begin{cases} 1 & \text{if } (\tilde{y}_j \geq 3 \wedge y_j \geq 3) \vee (\tilde{y}_j < 3 \wedge y_j < 3) \\ 0 & \text{otherwise} \end{cases} \tag{6}$$

Finally, a simple Format Reward ($R_{\text{format}}$) ensures the output adheres to the required structure, which consists of a binary judgment ("yes" or "no") followed by the numerical score in parentheses.

**Stage II: Fine-Grained Semantic Distinction.** Building on the foundational scoring abilities from Stage I, Stage II refines the model's capacity for nuanced semantic distinctions. This is achieved through a hybrid reward signal that incorporates both pairwise and listwise ranking objectives, both of which are enabled by our proposed Parallel Slice Ranking Reward (PSRR) mechanism.

PARALLEL SLICE RANKING REWARD MECHANISM. A primary challenge in directly optimizing ranking metrics is that a single, batch-level reward signal is too coarse to assign credit precisely. To address this, we propose the Parallel Slice Ranking Reward (PSRR) mechanism. The core idea of PSRR is to restructure the generated outputs to enable more granular reward computation. For a batch of $N$ samples, we begin by having the policy generate $G$ completions $\{o_{i,1}, \ldots, o_{i,G}\}$ for each sample. From each completion $o_{i,j}$, we then parse the predicted score $\tilde{y}_{i,j}$. Instead of treating the completions as a single flat list, we organize these $N \times G$ predicted scores into $G$ "parallel slices". Each slice, denoted as $Y_{\text{pred}}^j$, is defined as the collection of the $j$-th predicted score from all $N$ samples in the batch: $Y_{\text{pred}}^j = \{\tilde{y}_{1,j}, \tilde{y}_{2,j}, \ldots, \tilde{y}_{N,j}\}$, where $j \in \{1, \ldots, G\}$. This slicing architecture forms the foundation of our ranking rewards, ensuring each completion receives a specific learning signal based on its relative performance within its independent slice.

A sufficiently large batch size $N$ is crucial for a stable and meaningful ranking signal. To make this computationally feasible with limited GPU memory, we employ gradient accumulation strategy. Specifically, we first generate the full set of $N \times G$ completions and organize them into parallel slices to compute rewards and advantages globally. Subsequently, we process smaller sub-batches sequentially to compute losses and accumulate gradients over multiple backward passes before executing a single optimizer step. This strategy enables the model to learn from a rich ranking signal without incurring prohibitive memory costs.

PAIRWISE RANKING REWARD. Computed within each parallel slice $Y_{\text{pred}}^j$, this reward leverages the paired structure of the C-STS dataset to provide a local ranking signal. It is applied only to adjacent input samples that form a valid pair. We define the predicted difference as $\Delta_{\text{pred}} = \tilde{y}_{i,j} - \tilde{y}_{i+1,j}$ and the true difference as $\Delta_{\text{true}} = y_i - y_{i+1}$. To accommodate cases where paired samples share identical ground-truth scores ($\Delta_{\text{true}} = 0$), we distinguish them from strictly ordered pairs. The reward $R_{i,j}^{\text{pairwise}}$ is formulated as:

$$R_{i,j}^{\text{pairwise}} = \begin{cases} 1 - \frac{|\Delta_{\text{pred}}|}{D_{\max}} & \text{if } \Delta_{\text{true}} = 0 \\ R_{\text{base}} + (1 - R_{\text{base}}) \cdot \left(1 - \frac{|\Delta_{\text{pred}} - \Delta_{\text{true}}|}{D_{\max}}\right) & \text{if } \text{sgn}(\Delta_{\text{pred}}) = \text{sgn}(\Delta_{\text{true}}) \\ 0 & \text{otherwise} \end{cases} \quad (7)$$

where $D_{\max}$ represents the maximum possible score difference (i.e., 4 for the 1-5 scale). For strictly ordered pairs ($\Delta_{\text{true}} \neq 0$), the model receives a base reward $R_{\text{base}}$ for correct ordinality, with an additional penalty for prediction error magnitude. For tied pairs ($\Delta_{\text{true}} = 0$), the reward simply minimizes the absolute prediction difference.

LISTWISE RANKING REWARD. While the pairwise reward focuses on local comparisons, the list-wise reward provides a more global ranking perspective within each slice. It is calculated as the normalized difference between a completion's predicted rank and its ground-truth rank:

$$R_{i,j}^{\text{listwise}} = 1 - \frac{|\text{Rank}(\tilde{y}_{i,j}, Y_{\text{pred}}^j) - \text{Rank}(y_i, Y_{\text{true}})|}{N - 1} \quad (8)$$

where $Y_{\text{true}} = \{y_1, \ldots, y_N\}$ is the set of ground-truth labels for the current batch, the function $\text{Rank}(v, S)$ returns the rank of element $v$ within the set $S$ (from 1 to $N$), and the division by $N - 1$ normalizes the rank error to the range $[0, 1]$.

The final reward for Stage II, $R_{S2}$, combines the robust Pointwise Reward from Stage I as an anchor with the ranking-based rewards enabled by PSRR. The total reward is a weighted combination of these three components:

$$R_{S2} = \mu_1 R_{\text{pointwise}} + \mu_2 R_{\text{pairwise}} + \mu_3 R_{\text{listwise}} \quad (9)$$

## 3 EXPERIMENTS

### 3.1 EXPERIMENTAL SETUP

We build PoLi-RL upon the Qwen3 (Yang et al., 2025) model family (0.6B, 4B, and 8B) using the *ms-swift* framework (Zhao et al., 2024) for RL training. Detailed hyperparameter configurations are provided in Appendix A.3. All experiments are conducted on the official C-STS dataset (Deshpande et al., 2023). Following prior work, we use Spearman correlation as the primary metric and Pearson as the secondary. We compare our method against three baseline categories: First, discriminative models in the cross-encoder setting, such as SimCSE (Gao et al., 2022). Second, state-of-the-art proprietary and open-source reasoning models, including Deepseek-R1 (Guo et al., 2025), GPT-4 (Achiam et al., 2023), GPT-4o, Flan-T5 (Chung et al., 2024), Tk-Instruct (Wang et al., 2022), etc. Finally, our own SFT and few-shot implementations on Qwen3 backbones for direct comparison.

### 3.2 MAIN RESULTS

Table 1 summarizes the performance of PoLi-RL. Our framework establishes a new state-of-the-art (SOTA) for the cross-encoder architecture, with the 8B model achieving a Spearman correlation of **48.18**. The significance of this achievement is underscored by several key observations:

**Superiority over Strong Baselines.** PoLi-RL significantly outperforms the previous cross-encoder SOTA, SEAVER (Li et al., 2024), by a margin of **4.35** points. Compared to standard SFT, our method yields consistent gains across all model sizes. Notably, on Qwen3-0.6B, it yields substantial absolute improvements of **19.09** points over few-shot inference and **8.75** points over standard SFT, showcasing the substantial benefits of our progressive, multi-component reward optimization.

**Efficacy Across Model Scales.** A striking finding is the efficacy of PoLi-RL on smaller architectures. As shown in the results for Qwen3-0.6B and 4B, our framework enables compact models to perform significantly beyond their scale. Remarkably, our 0.6B model (44.34) not only outperforms the proprietary giant GPT-4 (43.6) but also surpasses the previous SOTA SEAVER (43.83). This demonstrates that the performance gains stem from aligning the reasoning process with ranking objectives, rather than relying solely on parameter scaling.

**Advantage over Proprietary Models.** Benchmarking against state-of-the-art reasoning models, our 8B model outperforms GPT-4o by 3.95 points and DeepSeek-R1 by 5.33 points. This suggests that while proprietary models possess strong general reasoning capabilities, they struggle to strictly align with the fine-grained quantization standards of C-STS in a few-shot setting. PoLi-RL bridges this gap by explicitly optimizing this alignment via RL, proving that a specialized, smaller model can surpass general-purpose giants on complex conditional ranking tasks.

Beyond standard benchmarks, we conduct rigorous validation to ensure the robustness of our claims. Comparisons with differentiable ranking objectives (Appendix A.4) confirm the superiority of RL over regression-based approaches. Furthermore, evaluation on the re-annotated C-STS dataset (Appendix A.5) validates PoLi-RL's SOTA performance, demonstrating that its gains stem from genuine reasoning improvements rather than overfitting to label noise.

Table 1: Main results on the official C-STS benchmark. All scores are reported as Spearman/Pearson correlation coefficients multiplied by 100. Results marked with † are obtained from (Deshpande et al., 2023), while ‡ denotes results from (Li et al., 2024).

| Methods | Training Paradigm | Parameters | Spearman ↑ | Pearson ↑ |
|---|---|---|---|---|
| *Discriminative Model Baselines (Cross-Encoder Architecture)* | | | | |
| RoBERTa$_{LARGE}$† | SFT | 355M | 40.7 | 40.8 |
| SimCSE$_{LARGE}$† | SFT | 355M | 43.2 | 43.2 |
| SEAVER SimCSE$_{LARGE}$‡ | SFT | 355M | 43.83 | 43.81 |
| *Generative Large Language Model Baselines* | | | | |
| Flan-T5$_{XXL}$† | Few-shot | 11B | 30.6 | - |
| Flan-UL2† | Few-shot | 20B | 23.5 | - |
| Tk-Instruct† | Few-shot | 11B | 17.8 | - |
| GPT-3.5† | Few-shot | 175B | 15.3 | - |
| DeepSeek-R1 | Few-shot | - | 42.85 | 42.36 |
| GPT-4† | Few-shot | - | 43.6 | - |
| GPT-4o | Few-shot | - | 44.23 | 44.07 |
| *Our Implementation on Qwen3-0.6B* | | | | |
| Qwen3-0.6B | Few-shot | 0.6B | 25.25 | 25.19 |
| Qwen3-0.6B | SFT | 0.6B | 35.59 | 36.83 |
| PoLi-RL (Qwen3-0.6B) | RL | 0.6B | **44.34** | **44.36** |
| *Our Implementation on Qwen3-4B* | | | | |
| Qwen3-4B | Few-shot | 4B | 37.97 | 38.48 |
| Qwen3-4B | SFT | 4B | 38.41 | 39.45 |
| PoLi-RL (Qwen3-4B) | RL | 4B | **46.23** | **46.19** |
| *Our Implementation on Qwen3-8B* | | | | |
| Qwen3-8B | Few-shot | 8B | 37.9 | 38.54 |
| Qwen3-8B | SFT | 8B | 40.42 | 40.83 |
| PoLi-RL (Qwen3-8B) | RL | 8B | **48.18** | **48.27** |

## 3.3 ABLATION STUDIES

**Effectiveness of the Two-Stage Curriculum and Reward Components.** Table 2 dissects the effectiveness of our progressive training schedule. We observe that a Naive RL approach (Row 2), which relies solely on a batch-level listwise objective, yields negligible improvement over the few-shot baseline (Row 1), highlighting the necessity of the progressive curriculum. Our PoLi-RL Stage

I (Row 3) addresses this by building a robust foundation, substantially outperforming the few-shot baseline by 6.87 points. Within this stage, ablating the binary reward (Row 4) leads to a discernible dip in performance, validating its role in grounding the model in the task's basic binary judgment.

Building upon this, the full PoLi-RL model (Row 5) further boosts performance by another 3.41 points (Row 5). Deconstructing the success of this final stage reveals that both ranking signals are vital: removing the listwise reward (Row 6) incurs the most significant penalty, while removing the pairwise reward (Row 7) also hinders performance. These findings confirm that both the two-stage curriculum and each of its constituent reward signals are essential for achieving optimal results, with the listwise signal being the most critical component in the final refinement stage.

Table 2: Ablation study on PoLi-RL's two-stage training design and its reward components. The $\Delta$ column shows the absolute improvement in Spearman correlation over the indicated baseline.

| Method | Reward Component(s) | Spearman ↑ | Pearson ↑ | $\Delta$ (Spearman) |
|---|---|---|---|---|
| (1) Few-shot Inference | - | 37.9 | 38.54 | - |
| (2) Naive RL | Listwise | 38.19 | 38.34 | +0.29 vs. (1) |
| (3) PoLi-RL (Stage I) | Pointwise + Binary | 44.77 | 44.45 | **+6.87** vs. (1) |
| (4)   - w/o Binary | Pointwise | 44.19 | 43.54 | -0.58 vs. (3) |
| (5) PoLi-RL (Full) | Pointwise + Pairwise + Listwise | **48.18** | **48.27** | **+3.41** vs. (3) |
| (6)   - w/o Listwise | Pointwise + Pairwise | 46.71 | 46.37 | -1.47 vs. (5) |
| (7)   - w/o Pairwise | Pointwise + Listwise | 47.6 | 47.59 | -0.58 vs. (5) |

**Sensitivity to Hyperparameters.**   We analyze the framework's robustness to key hyperparameters across both training stages. Regarding reward weights, Table 3 shows that Stage II is robust to weight variations, with peak performance achieved by moderately increasing the pairwise weight to 1.5. The model performs strongly even with the default equidistant weights (1:1:1), and shows considerable tolerance to pairwise signal, as halving its weight to 0.5 results in only a marginal performance drop to 47.77. Similarly, for the pointwise and listwise weights, deviations from their baseline of 1.0 result in only minor fluctuations. Crucially, the framework exhibits stable convergence across all configurations without training collapse. This confirms that our hybrid reward design effectively resolves the optimization difficulties encountered by naive listwise approaches.

Table 3: Ablation study on the reward weights ($\mu_1, \mu_2, \mu_3$) in PoLi-RL's Stage II.

| Method | $\mu_1$ (Pointwise) | $\mu_2$ (Pairwise) | $\mu_3$ (Listwise) | Spearman ↑ | Pearson ↑ |
|---|---|---|---|---|---|
| | 1.0 | 1.0 | 1.0 | 47.83 | 47.83 |
| | 1.0 | 1.5 | 1.0 | **48.18** | **48.27** |
| PoLi-RL | 1.5 | 1.0 | 1.0 | 47.3 | 47.23 |
| (Stage II) | 1.0 | 1.0 | 1.5 | 47.46 | 47.48 |
| | 1.0 | 0.5 | 1.0 | 47.77 | 47.31 |
| | 0.5 | 1.0 | 1.0 | 47.36 | 47.18 |
| | 1.0 | 1.0 | 0.5 | 47.39 | 47.27 |

To determine the optimal configuration for our PSRR mechanism, we study the impact of the slice size $N$, with results presented in Table 4. The empirical results reveal a clear trend: performance peaks at an intermediate slice size of $N = 24$ and degrades as the size deviates in either direction. This suggests that an optimal balance is required for the ranking signal. A slice that is too small may provide a less stable ranking signal, while one that is too large makes the ranking task overly complex for the model to learn effectively. This finding validates the design principle behind our PSRR mechanism: a carefully-sized, localized ranking signal is more effective than a purely global or an overly-restricted one. Further sensitivity analyses on the binary reward weights, pairwise base reward ($R_{\text{base}}$), and generation multiplicity ($G$) are provided in Appendix A.3, consistently validating the robustness of our default configurations.

Table 4: Analysis on the impact of the parallel slice size ($N$) in Stage II. $N$ represents the number of samples used for listwise ranking reward computation in each slice.

| Method | $N$ (Slice Size) | Spearman ↑ | Pearson ↑ |
|---|---|---|---|
| | 16 | 47.16 | 46.96 |
| | 24 | **48.18** | **48.27** |
| PoLi-RL(Stage II) | 32 | 47.44 | 47.19 |
| | 40 | 47.18 | 47.32 |
| | 48 | 46.78 | 46.84 |

## 3.4 GENERALIZABILITY OF PSRR MECHANISM

To empirically validate the generalizability of our framework beyond the C-STS domain, we extend our evaluation of PoLi-RL to the WMT-QE 2020 task (Fomicheva et al., 2020). While this task shares the overarching goal of optimizing global Spearman correlation, it diverges fundamentally from C-STS across three dimensions: (1) Domain: multilingual translation quality estimation, (2) Label Scale: continuous scores from 0 to 100, and (3) Data Structure: independent samples lacking the deterministic paired ordinality of C-STS.

For this experiment, we employ a minimalist configuration of PoLi-RL by disabling the C-STS-specific pairwise and binary rewards. We relied solely on the pointwise reward (Stage I) and the PSRR-based listwise ranking reward (Stage II), with prompt settings adapted from Sato et al. (2024).

As shown in Table 5, even with this simplified setup, PoLi-RL achieves a Spearman correlation of **54.33**, yielding a substantial **3.43** point gain over the strong SFT baseline (50.90). Furthermore, the progression from Stage I (51.72) to Stage II explicitly highlights the added value of the listwise ranking objective. These results confirm that the PSRR mechanism is not merely a task-specific optimization for C-STS, but a transferable solution for aligning LLMs with listwise ranking objectives across domains and label scales.

Table 5: Generalization performance on WMT-QE 2020 task (en-zh subset) using only the core PSRR mechanism, excluding C-STS-specific reward components.

| Methods | Training Paradigm | Spearman ↑ | Pearson ↑ |
|---|---|---|---|
| Qwen3-8B | Few-shot | 45.03 | 44.18 |
| Qwen3-8B | SFT | 50.90 | **51.09** |
| PoLi-RL (Stage I) | RL | 51.72 | 50.58 |
| PoLi-RL (Stage II) | RL | **54.33** | **51.09** |

## 4 ANALYSIS

### 4.1 ANALYSIS OF PREDICTION ERROR DISTRIBUTION

Figure 3 visualizes the distribution of absolute prediction errors on the C-STS validation set. The plot reveals several key insights into the models' behaviors. While an error of 1 is the most frequent outcome for all models, likely reflecting the inherent nuances of the C-STS scale, a clear progression of improvement is evident. Compared to the few-shot and SFT model, PoLi-RL demonstrates a more favorable error distribution. It not only achieves the highest density of perfect predictions (error=0) but, more importantly, effectively suppresses the probability of high-error predictions (errors $\geq 2$). This indicates that our method significantly reduces the frequency of large, unreliable deviations, yielding a model that is more precise against severe misjudgments.

### 4.2 QUALITATIVE ANALYSIS: A CASE STUDY ON NUANCED REASONING

To qualitatively analyze our framework's nuanced reasoning ability, we present a case study in Figure 4 on the challenging condition "The person's connection with the ground". The baseline models show clear deficiencies: the few-shot model exhibits a brittle, literal interpretation, fixating

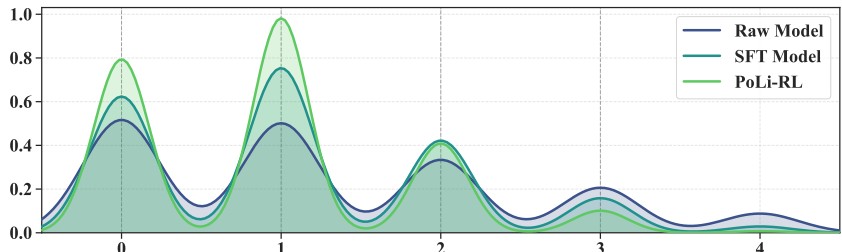

Figure 3: Distribution of absolute prediction errors for the raw model, SFT model, and PoLi-RL. The $x$-axis represents the absolute error ($|\text{Predicted} - \text{True}|$), and the $y$-axis is the probability density.

on the preposition 'on' while ignoring the physical context, leading to an incorrect prediction. The SFT Baseline, while closer to the target, lacks the reasoning depth to make the necessary fine-grained distinction. In contrast, PoLi-RL applies world knowledge to infer that both "jumping" and "doing a flip" result in the same "not connected" state, arriving at the accurate score of 4 by identifying the differing actions leading to the state as unimportant details. This case study demonstrates that PoLi-RL cultivates the contextual inference ability required for complex conditional judgments.

---

**Case Study: Model Comparison on a Reasoning Sample**

**Sentence 1:** A person riding a snowboard jumps high over the snowy hill.
**Sentence 2:** A man in a blue snowsuit does a flip on a huge snowy mountain.
**Condition:** The person's connection with the ground.
**Gold Label:** 4 (Mostly equivalent)

- - - - - - - - - - - - - - - - - - - - - - - - - - - - - - - - - - - - - - - - - - - - -

**Few-shot Inference Prediction:** no(1)
**Reasoning:** *In Sentence 1... , the key action is jumping, which implies the person is airborne and not in contact with the ground. In Sentence 2... , the preposition 'on' suggest that he's on the mountain's surface. So, he's connected to the ground. Therefore, the two sentences have different connections with the ground. Hence, the answer is 'no(1)'.*

- - - - - - - - - - - - - - - - - - - - - - - - - - - - - - - - - - - - - - - - - - - - -

**SFT Baseline Prediction:** yes(3)

- - - - - - - - - - - - - - - - - - - - - - - - - - - - - - - - - - - - - - - - - - - - -

**PoLi-RL Prediction:** yes(4)
**Reasoning:** *Sentence 1 describes... . So he's airborne during the jump, meaning he's not connected to the ground. Sentence 2 talks about ... I infer that this action, similar to a jump or a trick in a snow-sports context, also means he is in the air. Therefore, they are similar because both have the person not connected to the ground. But the final score should be 4 because the action (jumping/flipping) leading to this state represent minor, differing details.*

---

Figure 4: A detailed case study comparing the reasoning processes and final outputs of the Few-shot, SFT baselines, and PoLi-RL on a C-STS sample that requires nuanced reasoning.

## 5  CONCLUSION

We introduce PoLi-RL, a two-stage reinforcement learning framework that resolves the coarse-grained credit assignment problem associated with listwise objectives in C-STS through a progressive curriculum and an innovative Parallel-Slice Ranking Reward (PSRR) mechanism. Our method establishes a new state-of-the-art for the cross-encoder architecture, significantly outperforming SFT baselines and even proprietary reasoning models like GPT-4o. As the first successful application of reinforcement learning to this task, our study validates a powerful paradigm for aligning LLMs with complex ranking objectives, demonstrating the framework's potential to other ranking-based tasks.

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

# A APPENDIX

## A.1 RELATED WORK

**Conditional Semantic Textual Similarity.** C-STS is a recent advancement over traditional STS (Nakov et al., 2013; Zhang et al., 2024a; 2025b) that introduces a natural language condition to disambiguate the measurement of similarity between two texts. Research in this area has primarily established three mainstream paradigms: Bi-encoder, Tri-encoder, and Cross-encoder. Given two texts and a condition, the bi-encoder architecture typically uses a Siamese network to encode two text-condition pairs, while the tri-encoder architecture encodes each text and the condition separately before an aggregation step. A prevalent optimization strategy for these paradigms is contrastive learning. For instance, (Liu et al., 2025) propose a conditional contrastive learning framework that pulls representations of the same text pair under a high-similarity condition closer, while pushing them apart under a low-similarity one. Extending this, Hyper-CL (Yoo et al., 2024) utilizes a tri-encoder setup where a hypernetwork generates condition-specific projectors to dynamically adapt sentence representations within a contrastive objective. More recently, CSR introduced a parameter-free router for the tri-encoder, using the condition to re-weight sentence tokens to amplify relevant information without increasing model size.

In contrast, the cross-encoder architecture processes the concatenated texts and condition as a single input, enabling deep, token-level interaction. However, this theoretical advantage did not consistently translate to superior performance in earlier discriminative models. The state-of-the-art method in this setting, SEAVER, addressed this discrepancy by identifying that such models can be distracted by condition-irrelevant tokens. To resolve this, SEAVER (Li et al., 2024) introduces an attention reallocation mechanism that optimizes the model by re-weighting internal attention scores during fine-tuning, forcing a focus on the most salient information.

The advent of LLMs has introduced new approaches, mainly few-shot inference and using LLMs as feature extractors (Yamada & Zhang, 2025) (Li & Li, 2023). However, all these prior works, regardless of architecture, are confined to supervised paradigms like SFT or contrastive learning. Our work is the first to apply reinforcement learning to this task.

**Reinforcement Learning for Large Language Models.** Reinforcement Learning (RL) is pivotal for aligning and enhancing Large Language Models (LLMs), with algorithms like Proximal Policy Optimization (PPO) (Schulman et al., 2017) widely used to optimize non-differentiable objectives in tasks such as reasoning and code generation. To address PPO's limitations, such as high variance in long-sequence tasks, advanced variants have emerged.

Group Relative Policy Optimization (GRPO) (Guo et al., 2025) addresses these limitations by introducing a group-relative advantage estimation, which eliminates the need for a separate value function through Z-score normalization of rewards within sample groups. This approach simplifies training and enhances sample efficiency, as demonstrated in models like DeepSeek-R1. Building on this, Decoupled Clip and Dynamic Sampling Policy Optimization (DAPO) (Yu et al., 2025) provides an open-source, scalable RL system tailored for LLMs. DAPO incorporates key improvements, including dynamic sampling to adaptively adjust the number of generated completions based on reward variance, making it particularly effective for long-horizon reasoning tasks.

Leveraging DAPO's powerful optimization engine, our work, PoLi-RL, marks a significant departure by being the first to introduce a reinforcement learning framework to the C-STS task, thereby establishing a new optimization paradigm.

## A.2 PROMPT TEMPLATE FOR POLI-RL

Below is the detailed few-shot prompt used for both the few-shot inference baseline and the training, evaluation process of PoLi-RL.

**Prompt for C-STS Task**

Judge the semantic similarity between Sentence 1 and Sentence 2 based **completely** on the given Condition. The final output must be exactly in this format: the similarity judgment ('yes' or 'no') followed by the score in parentheses, wrapped in `<answer></answer>` tags. Examples: `<answer>yes(4)</answer>`, `<answer>no(1)</answer>`. Include no other text, tags, or explanations.

To arrive at this output, follow these two steps:

**Step 1: Binary Judgment.** Determine if the sentences are 'similar' ('yes') or 'not similar' ('no').

- 'similar': The sentences are roughly, mostly, or completely equivalent under the condition.
- 'not similar': The sentences are dissimilar under the condition.

**Step 2: Fine-grained Score.** Assign an integer score based on Step 1:

- For a 'yes' judgment:
  * **5:** The two sentences are completely equivalent as they mean the same thing with respect to the condition.
  * **4:** The two sentences are mostly equivalent, but some unimportant details differ with respect to the condition.
  * **3:** The two sentences are roughly equivalent, but some important information differs or is missing with respect to the condition.
- For a 'no' judgment:
  * **2:** The two sentences are dissimilar, but are on a similar topic with respect to the condition or shares a close semantic relationship. This applies when items are clearly different, but not direct opposites.
  * **1:** The two sentences are dissimilar with respect to the condition, representing a direct opposition or a clear, unrelated difference. (e.g., 'man' vs. 'woman').

Here are some examples:

- - - - - - - - - - - - - - - - - - - - - - - - - - - - - - - - - - - - - - - - - - - - - - - - - - - -

**Example 1:**
`<Sentence1>`: A girl is cooking in a kitchen and a man is standing next to her.
`<Sentence2>`: A man sitting with a pizza in his hand in front of pizza on the table.
`<Condition>`: The number of people.
`<answer>no(1)</answer>`
*Explanation: The first sentence mentions two people, while the second sentence mentions only one person.*

- - - - - - - - - - - - - - - - - - - - - - - - - - - - - - - - - - - - - - - - - - - - - - - - - - - -

**Example 2:**
`<Sentence1>`: A wood table sitting by a wood framed bed with a lamp on it.
`<Sentence2>`: A microwave, refrigerator, television, and wooden drawers sit in the corner of a bedroom.
`<Condition>`: The room type.
`<answer>yes(5)</answer>`
*Explanation: We can infer from the two sentences that the room type are both bedroom.*

- - - - - - - - - - - - - - - - - - - - - - - - - - - - - - - - - - - - - - - - - - - - - - - - - - - -

**Example 3:**
`<Sentence1>`: A small crowd gathered around the injured person.
`<Sentence2>`: A crowd jumps up and down to the tunes played by an artist.
`<Condition>`: The number of people
`<answer>yes(3)</answer>`
*Explanation: While both sentences mention crowds, it is important and unclear how many people there are.*

- - - - - - - - - - - - - - - - - - - - - - - - - - - - - - - - - - - - - - - - - - - - - - - - - - - -

Now, apply these steps to the following sentences:

`<Sentence1>`: {sentence1}
`<Sentence2>`: {sentence2}
`<Condition>`: {condition}

A.3 HYPERPARAMETER CONFIGURATIONS AND OTHER ABLATION STUDY

This section details the hyperparameter settings and provides empirical justification for selections not covered in the main paper. The default configurations used for our main results are summarized in Table 6.

Table 6: Default hyperparameter configurations for the main results.

| HyperParameter | Default Value |
|---|---|
| $\lambda_1 : \lambda_2 : \lambda_3$ | 1:0.25:1 |
| $\mu_1 : \mu_2 : \mu_3$ | 1:1.5:1 |
| $R_{\text{base}}$ | 0.5 |
| $G$ | 8 |
| $N$ | 24 |

In addition to the ablation studies presented in the main paper, we conducted sensitivity analyses on three key hyperparameters: the binary reward weights in Stage I, generation multiplicity ($G$) and the pairwise base reward ($R_{\text{base}}$).

**Sensitivity to Reward Weights in PoLi-RL Stage I.** In Stage I, the Binary Judgment Reward serves as an auxiliary signal to prevent the model from converging to "safe" median scores. We test the sensitivity to the weight of this component ($\lambda_2$). Results in Table 7 show that PoLi-RL is highly robust to variations in $\lambda_2$, consistently establishing a strong foundation for Stage II training.

Table 7: Sensitivity to Binary Reward Weight ($\lambda_2$) in PoLi-RL's Stage I.

| Method | $\lambda_1$ (Pointwise) | $\lambda_2$ (Binary) | $\lambda_3$ (Format) | Spearman ↑ | Pearson ↑ |
|---|---|---|---|---|---|
| | 1.0 | 0.1 | 0.1 | **44.94** | **44.93** |
| PoLi-RL | 1.0 | 0.25 | 0.1 | 44.77 | 44.45 |
| (Stage I) | 1.0 | 0.5 | 0.1 | 44.67 | 44.42 |
| | 1.0 | 1.0 | 0.1 | 44.76 | 44.92 |

**Impact of Generation Multiplicity.** We investigate the trade-off between exploration diversity and computational overhead by varying $G \in \{4, 8, 12\}$. As shown in Table 8, while increasing $G$ theoretically aids RL exploration, empirical results indicate that $G = 4$ offers insufficient exploration coverage: while the restricted search space may suffice for basic scoring, it fails to generate the diverse candidates necessary for learning the fine-grained semantic distinctions in Stage II. Conversely, scaling to $G = 12$ leads to performance saturation and incurs higher computational overhead. These results validate our selection of $G = 8$ as the optimal configuration for balancing performance and efficiency.

Table 8: Ablation on generation multiplicity ($G$) in PoLi-RL both stages.

| $G$ | Method | Spearman ↑ | Pearson ↑ |
|---|---|---|---|
| 4 | PoLi-RL(Stage I) | 44.55 | 44.49 |
| 4 | PoLi-RL(Stage II) | 47.47 | 47.45 |
| 8 | PoLi-RL(Stage I) | 44.77 | 44.45 |
| 8 | PoLi-RL(Stage II) | **48.18** | **48.27** |
| 12 | PoLi-RL(Stage I) | 45.51 | 45.61 |
| 12 | PoLi-RL(Stage II) | 47.98 | 48.06 |

**Rationale for Pairwise Base Reward.** The pairwise reward (Eq. 7) incorporates a constant base reward $R_{base}$ to ensure a guaranteed positive signal when the model correctly predicts the ranking direction (ordinality), even if the exact score gap (cardinality) is imprecise. We compare the default

$R_{base} = 0.5$ against removing it (0.0) or overweighting it (0.75). Table 9 demonstrates that 0.5 yields the best performance, justifying the need for a balanced reward structure.

Table 9: Sensitivity analysis of the pairwise base reward ($R_{base}$).

| Method | $R_{base}$ | Spearman ↑ | Pearson ↑ |
|---|---|---|---|
| | 0 | 47.90 | 47.59 |
| | 0.25 | 47.78 | 47.81 |
| PoLi-RL (Stage II) | 0.5 | **48.18** | **48.27** |
| | 0.75 | 47.54 | 47.33 |

### A.4 COMPARISON WITH DIFFERENTIABLE RANKING OBJECTIVES

In this section, we explicitly compare PoLi-RL against a strong cross-encoder regression baseline trained with differentiable ranking objectives. The goal is to determine whether optimizing a surrogate loss is sufficient to capture the rank-based nuances of C-STS. Specifically, we compare PoLi-RL against a baseline trained with the Pearson Correlation Coefficient (Pcc) Loss (Zhang & Li, 2024a), a state-of-the-art differentiable proxy for Spearman metric. Pcc loss is defined as:

$$\mathcal{L} = 1 - \frac{cov(X, Y)}{\sigma_X \sigma_Y} \tag{10}$$

where $X$ represents the predicted scores and $Y$ is the ground-truth labels. Table 10 shows that while the regression baseline outperforms standard SFT, PoLi-RL still maintains a clear advantage (1.74). This superiority stems from the paradigm shift: Regression treats the model as a "black box" that maps embeddings to a scalar. In contrast, PoLi-RL optimizes the reasoning process itself. By aligning the token-level generation probability with the non-differentiable ranking metric, the model learns how to reason towards the correct score, enabling better generalization in complex conditional scenarios.

Table 10: Comparison between PoLi-RL and SFT with differentiable Pearson Correlation Coefficient (Pcc) Loss regression.

| Methods | Training Paradigm | Spearman ↑ | Pearson ↑ |
|---|---|---|---|
| Qwen3-8B | Few-shot | 37.90 | 38.54 |
| Qwen3-8B | SFT (Auto-regressive) | 40.42 | 40.83 |
| Qwen3-8B | SFT (Regression) | 46.44 | 46.59 |
| PoLi-RL (Qwen3-8B) | RL | **48.18** | **48.27** |

### A.5 PERFORMANCE ON RE-ANNOTATED C-STS DATASET

Recent work by Tu et al. (2024) identified potential label noise in the original C-STS dataset and released a re-annotated validation set. Subsequently, Zhang et al. (2025c) further refined the dataset by correcting the condition descriptions and utilizing LLMs to clean the training labels. To ensure the robustness of our method against data quality issues, we re-evaluate PoLi-RL on the refined dataset provided by Zhang et al. (2025c).

As detailed in Table 11, PoLi-RL maintains its significant performance advantage on the re-annotated data, achieving a state-of-the-art Spearman correlation of 76.08 and outperforming the SFT baseline by nearly 4 points. This confirms that our reported improvements are robust and valid, rather than an artifact of overfitting to label noise.

### A.6 LLM USAGE STATEMENT

The large language model (LLM) was utilized during the preparation of this manuscript. The use of this technology was strictly confined to the role of a writing assistant for the sole purpose of

Table 11: Performance on the Re-annotated C-STS Dataset. Evaluation performed on the 30% hold-out split of the validation set following the protocol of Tu et al. (2024) and Zhang et al. (2025c). Results marked with † are obtained from (Tu et al., 2024).

| Methods | Training Paradigm | Parameters | Spearman ↑ | Pearson ↑ |
|---|---|---|---|---|
| QA-SimCSE$_{\text{LARGE}}$† | SFT | 355M | 72.9 | 72.3 |
| GPT-3.5† | Few-shot | - | 66.1 | 64.8 |
| GPT-4† | Few-shot | - | 64.4 | 60.2 |
| Qwen3-8B | Few-shot | 8B | 64.42 | 64.5 |
| Qwen3-8B | SFT | 8B | 72.09 | 70.41 |
| PoLi-RL (Qwen3-8B) | RL | 8B | **76.08** | **74.16** |

improving the linguistic quality of the text. Specifically, the LLM was employed for tasks related to grammar, syntax, phrasing, and overall readability. Its function was exclusively to perform surface-level linguistic refinements on text already written by the human authors. Crucially, the LLM did not contribute to any substantive or intellectual aspects of the research. The conceptualization of the study, the design of the methodology, the execution of experiments, the interpretation of results, and the formulation of conclusions were all executed by the human authors.

