# OpenReview forum: "PoLi-RL: A Point-to-List Reinforcement Learning Framework for Conditional Semantic Textual Similarity"
_ICLR.cc/2026/Conference — ICLR 2026 Poster_

### Official Review · Reviewer_6HRs · 2025-10-29

**Soundness:** 3
**Presentation:** 2
**Contribution:** 2
**Rating:** 6
**Confidence:** 4

**Summary:**

The paper proposes PoLi-RL, a two-stage point-to-list reinforcement learning framework for conditional STS with a cross-encoder backbone. Stage I teaches the model to “score” with pointwise distance, binary-consistency, and formatting rewards; Stage II adds pairwise and listwise ranking signals computed over parallel slices of completions (PSRR), yielding finer credit assignment aligned with rank-correlation objectives. Experiments on the standard C-STS benchmark (with an 8B-scale backbone) show consistent gains over few-shot prompting and supervised fine-tuning; ablations indicate that both the curriculum and the ranking-based rewards contribute.

**Strengths:**

- The slice-wise ranking mechanism produces differentiated signals per completion, better aligned to Spearman/Kendall-style goals than batch-level listwise rewards.
- The paper explains how to make PSRR tractable (generation multiplicity, gradient accumulation), offering a reproducible training recipe.
- Strong performance with an 8B model suggests good cost-effectiveness.

**Weaknesses:**

- Comparisons to the latest closed-source systems (e.g., GPT-4o, Claude-3.7 Sonnet) are missing, leaving the method’s headroom to stronger proprietary models underexplored.
- All results are on Qwen3-8B only. Please add multi-size backbones or report compute-parity/parameter-parity comparisons to strengthen fairness claims.
- The few-shot template and examples are relegated to the appendix; the main text’s notation does not formalize the few-shot setup, which hinders reproducibility. The authors should surface the key prompt template and define the few-shot variables in the notation section.
- C-STS has been shown to contain substantial label noise [1], and a recent study has produced a re-annotated version of the data [2]. Training and evaluating on this re-annotated dataset would provide a more accurate assessment of the model’s effectiveness. I understand that [2] was released only a few days before the ICLR deadline; however, given the importance of data quality, I believe that evaluation on the newly annotated data would more clearly demonstrate the actual performance gains.

[1] Linguistically conditioned semantic textual similarity. ACL2024

[2] Annotating Training Data for Conditional Semantic Textual Similarity Measurement using Large Language Models. EMNLP2025

**Questions:**

- How does performance scale with model size (e.g., 3B/14B/32B) and with generation multiplicity G and slice size N?
- Why RL over differentiable ranking losses? Under equal data/compute, how do listwise/pairwise differentiable surrogates perform relative to PoLi-RL Stage II?
- What is the exact hyperparameter-tuning protocol? Which hyperparameters were tuned on dev, what ranges were explored, and what metric decided the final selection?

---

> ### Author Response · Authors · 2025-11-21
>
> Thank you for the constructive feedback and recognizing the novelty of our slice-wise ranking mechanism. We have addressed all concerns with extensive new experiments.
>
> **Weaknesses 1:** We agree that benchmarking against SOTA proprietary models is essential. We have added evaluations for **GPT-4o** and **DeepSeek-R1** using the same few-shot setup (see Table below).
>
> | Method               | Training Paradigm | Spearman  |  Pearson  |
> | -------------------- | :---------------: | :-------: | :-------: |
> | GPT-4                |     Few-shot      |   43.60   |     -     |
> | DeepSeek-R1 (new)    |     Few-shot      |   42.85   |   43.36   |
> | GPT-4o (new)         |     Few-shot      |   44.23   |   44.07   |
> | PoLi-RL (Qwen3-0.6B) |        RL         |   44.34   |   44.36   |
> | PoLi-RL (Qwen3-4B)   |        RL         |   46.23   |   46.19   |
> | PoLi-RL (Qwen3-8B)   |        RL         | **48.18** | **48.27** |
>
> Our 8B model significantly outperforms **GPT-4o (+3.95)** and **DeepSeek-R1 (+5.33)**. Even more remarkably, our **0.6B** model slightly edges out these massive proprietary models. This suggests that while proprietary models possess strong general reasoning capabilities, they struggle to strictly align with the fine-grained quantization standards of C-STS (1-5 scale) in a few-shot setting. PoLi-RL bridges this gap by explicitly optimizing this alignment via RL, proving that a specialized, smaller model can surpass general-purpose giants on complex conditional ranking tasks.

---

> > ### Author Response · Authors · 2025-11-21
> >
> > **Weaknesses 2 & Question 1:**  We conducted a full set of new experiments using **Qwen3-0.6B** and **Qwen3-4B** models. We also performed ablations on generation multiplicity $G$.
> >
> > - Experiments on **Qwen3-0.6B** and **Qwen3-4B**:
> >
> >   | Method               | Training Paradigm | Spearman  |  Pearson  |
> >   | -------------------- | :---------------: | :-------: | :-------: |
> >   | Qwen3-0.6B           |     Few-shot      |   25.25   |   25.19   |
> >   | Qwen3-0.6B           |        SFT        |   35.59   |   36.83   |
> >   | PoLi-RL (Qwen3-0.6B) |        RL         |   44.34   |   44.36   |
> >   | Qwen3-4B             |     Few-shot      |   37.97   |   38.48   |
> >   | Qwen3-4B             |        SFT        |   38.41   |   39.45   |
> >   | PoLi-RL (Qwen3-0.6B) |        RL         | **46.23** | **46.19** |
> >
> >   The results turn out that PoLi-RL yields consistent gains across all sizes. Most notably, our 0.6B model (44.34) outperforms the previous Cross-Encoder SOTA (SEAVER: 43.83). This confirms that the performance gain stems from the PoLi-RL framework's effectiveness rather than just parameter scale.
> >
> > - Ablation studies on $G$: While larger $G$ theoretically aids RL exploration, empirical results indicate that $G=4$ lacks sufficient candidate diversity for the complex ranking in Stage II. Conversely, scaling to $G=12$ leads to performance saturation and incurs higher computational overhead. Thus, $G=8$ provides the necessary diversity for robust ranking without incurring excessive costs.
> >
> >   | G            |       Method       | Spearman  |  Pearson  |
> >   | ------------ | :----------------: | :-------: | :-------: |
> >   | 4 (new)      | PoLi-RL (Stage I)  |   44.55   |   44.49   |
> >   | 4 (new)      | PoLi-RL (Stage II) |   47.47   |   47.45   |
> >   | 8 (original) | PoLi-RL (Stage I)  |   44.77   |   44.45   |
> >   | 8 (original) | PoLi-RL (Stage II) | **48.18** | **48.27** |
> >   | 12 (new)     | PoLi-RL (Stage I)  |   45.51   |   45.61   |
> >   | 12 (new)     | PoLi-RL (Stage II) |   47.98   |   48.06   |
> >
> > - Ablation studies on $N$: We've already conducted an ablation study on $N$ at Table 4. The performance peaks at $N=24$. This suggests that an optimal balance is required for the ranking signal. Smaller slices ($N<16$) may provide noisy ranking signals, while larger slices ($N>40$) make the ranking task too difficult for the model to distinguish subtle differences effectively.

---

> > > ### Author Response · Authors · 2025-11-21
> > >
> > > **Weaknesses 3:** We appreciate you for pointing out this gap in reproducibility. We acknowledge that the notation $p=INS(x)$ in Eq. (1) oversimplified the input structure by omitting the few-shot demonstrations. In the revision, we will explicitly formalize the few-shot setup in the notation and surface the key prompt template to ensure full reproducibility.
> > >
> > > ---
> > >
> > > **Weaknesses 4:** Thank you for pointing out this vital new resource. Follow the default settings, we re-ran our core experiments on the newly released re-annotated C-STS dataset (EMNLP 2025).
> > >
> > > | Method             | Training Paradigm | Spearman  |  Pearson  |
> > > | ------------------ | :---------------: | :-------: | :-------: |
> > > | Qwen3-8B           |     Few-shot      |   64.42   |   64.50   |
> > > | Qwen3-8B           |        SFT        |   72.09   |   70.41   |
> > > | PoLi-RL (Stage I)  |        RL         |   74.74   |   73.49   |
> > > | PoLi-RL (Stage II) |        RL         | **76.08** | **74.16** |
> > >
> > > PoLi-RL achieves an higher absolute score on the cleaner dataset and maintains a robust lead over SFT (**+3.99** Spearman), proving our method's effectiveness is not an artifact of label noise.

---

> > > > ### Author Response · Authors · 2025-11-21
> > > >
> > > > **Question 2:** To rigorously address this, We formulated C-STS as a regression task and implemented a strong **Regression Baseline** using the **Pearson Correlation Coefficient (PCC) Loss**, which was recently proposed as a differentiable proxy for breaking the performance ceiling of contrastive learning [1].
> > > >
> > > > Pcc loss is defined as $\mathcal{L} = 1 - \frac{cov(X, Y)}{\sigma_X \sigma_Y}$ (where $X$ is the predicted score and $Y$ is the label) . It serves as a theoretically sound, listwise-aware differentiable proxy for the non-differentiable Spearman metric.
> > > >
> > > > | Method        |   Training Paradigm   | Spearman  |  Pearson  |
> > > > | ------------- | :-------------------: | :-------: | :-------: |
> > > > | Qwen3-8B      |       Few-shot        |   37.90   |   38.54   |
> > > > | Qwen3-8B      | SFT (Auto-regressive) |   40.42   |   40.83   |
> > > > | Qwen3-8B      |   SFT (Regression)    |   46.44   |   46.59   |
> > > > | PoLi-RL(Ours) |          RL           | **48.18** | **48.27** |
> > > >
> > > > While this Regression Baseline significantly outperforms standard SFT, **PoLi-RL still outperforms it by +1.74 points**. This superiority stems from the paradigm shift: Regression treats the model as a "black box" mapping embeddings to a scalar. In contrast, PoLi-RL does not just learn what the score is, but how to reason towards it. This alignment between the reasoning process and the final ranking metric allows the model to generalize better to complex conditional scenarios.
> > > >
> > > > [1] Pcc-tuning: Breaking the Contrastive Learning Ceiling in Semantic Textual Similarity. EMNLP 2024
> > > >
> > > > ---
> > > >
> > > > **Question 3:** We only tuned $\mu_1:\mu_2:\mu_3$ and slice size $N$ on the official validation set. For reward weights in stage II, we start with $\mu_1:\mu_2:\mu_3$=1:1:1, and found that slightly up-weighting the pairwise component (1.5) yielded better performance. For slice size, we tuned $N \in \{16, 24, 32, 40, 48\}$ and selected $N=24$ based on the peak Spearman score. All selections were based on the validation Spearman correlation.

---

> > > > > ### Author Response · Authors · 2025-11-27
> > > > > **Update: Revised Manuscript**
> > > > >
> > > > > Dear Reviewer 6HRs,
> > > > >
> > > > > Thank you again for your helpful suggestions. We have uploaded the **revised manuscript** to reflect the improvements made based on your review. Specifically, we have incorporated the following updates:
> > > > >
> > > > > - Hyperparameters: Ablation studies details in Appendix A.3.
> > > > > - Scalability (0.6B, 4B models): Detailed in Appendix A.4.
> > > > > - Reasoning Baselines: Added in Appendix A.5.
> > > > > - RL vs. Differentiable Ranking Loss: Comparison in Appendix A.6.
> > > > > - Performance on C-STS re-annotated data: Results in Appendix A.7.
> > > > > - Revised Section 2.1: Formally defined the few-shot variables and prompted setup to ensure reproducibility.
> > > > >
> > > > > We believe these revisions comprehensively address your concerns. We look forward to your thoughts on the updated paper.
> > > > >
> > > > > Best regards, Authors

---

### Official Review · Reviewer_PEjZ · 2025-10-30

**Soundness:** 2
**Presentation:** 3
**Contribution:** 2
**Rating:** 4
**Confidence:** 3

**Summary:**

The method is novel and promising to some extent, with respectable results on C-STS; however, evidence for generalization is nearly absent (the reward and training organization are over-specialized to C-STS). Combined with missing strong reasoning baselines and insufficient stabilization details, the present claims do not extrapolate well.

**Strengths:**

S1 Technical fit. Using RL to align with non-differentiable rank-correlation/ranking metrics (e.g., Spearman; more broadly NDCG, etc.) is naturally consistent with how C-STS is evaluated.
S2 Method design. The two-stage curriculum (Point → List) + PSRR’s “parallel slicing” refines credit assignment; the engineering gives reproducible slicing/normalization and a weighting composition.
S3 Competitive results. Achieves Spearman comparable to or better than strong existing baselines on the C-STS cross-encoder track, addressing the observation that “even strong models may not break 50” on this task.
S4 Reproducibility. Anonymous code and checkpoints are provided.

**Weaknesses:**

W1 Baseline coverage.
The paper only reports scores of non-reasoning large models (e.g., GPT-4) on C-STS, while reasoning models have progressed rapidly this year. The paper lacks strong baselines for **reasoning-style** models (e.g., DeepSeek-R1, OpenAI o3) under few-shot/CoT, leaving open the concern that “C-STS may already be easily solved by reasoning models.”
W2 Severe lack of generalizability.
There is essentially no evidence supporting “the method generalizes,” and the reward design is highly specialized to C-STS’s annotation and organization:
1. Stage-I strongly relies on the 1–5 score scale (Eq. (5) uses min-max normalization) and the ≥3 / ≤2 binary threshold (Eq. (6)).
2. Stage-II’s pairwise term applies only to “adjacent pairs,” normalized by “maximum difference = 3” (Eq. (7)), which is tightly bound to C-STS’s pairing construction and score gaps.
3. The listwise reward is built on PSRR’s cross-sample slices (Eq. (8)), rather than the more general definition of “ranking a candidate set per input/query.” This makes the choice of “list domain” heavily affected by batch and slicing, with limited extrapolative interpretability.
4. There is no validation across conditions/domains/data sources; no changes of label scales (e.g., 0–3, 0–4), no alternative list domains, nor robustness reports under different noise/shift settings.
Taken together, the current evidence only supports the weak claim that the method is “effective on C-STS.” Without systematic experiments across setups, the paper’s main claims lack external validity.
W3 Insufficient theoretical and empirical support for hyperparameters and reward design.
The paper introduces many reward components and hyperparameters (Stage-I’s R_pointwise, R_binary, R_format and weight (λ); Stage-II’s (R_pairwise, R_listwise), baseline constant (R_base ), slice/batch sizes (N, G), and “maximum difference = 3,” etc.), but offers only intuitive motivation. It lacks (i) testable theoretical grounds/invariance guarantees (why the shaping does not change the optimal policy, or why it aligns with the target metric), and (ii) systematic necessity evidence (component-wise/hyperparameter ablations and significance tests). Current experiments cover only a minority of hyperparameters; several key choices remain unexplained or unvalidated, making it hard to disentangle method gains from empirical tuning.

**Questions:**

Q1. Please add strong baselines for recent reasoning-style models under few-shot/CoT.
Q2. The current setup relies heavily on the 1–5 label scale, adjacent pairing, and a “cross-sample slice” list domain. Please provide any empirical evidence for generalizability:
1. Cross-condition/cross-domain/cross-source evaluation;
2. Changing label scales and thresholds (non-1–5, non-≥3/≤2);
3. Replace the list domain from “cross-sample slicing” to “per-input candidate set,” and compare stability/effectiveness;
4. Robustness under noise and distribution shift.
If you do not intend to generalize, explicitly delimit the applicability scope and tone down the claims in the abstract/conclusion.
Q3. Please provide testable rationale for “why these settings,” plus the full hyperparameter details and reasons. Explain the design motivation for each reward and its structure; for several key hyperparameters, add principled justifications or ablations.

---

> ### Author Response · Authors · 2025-11-21
>
> Thank you for the detailed and critical assessment. We value the opportunity to demonstrate the robustness and theoretical grounding of PoLi-RL. We have addressed the concerns regarding reasoning baselines, generalizability, and hyperparameter justification with extensive new experiments and theoretical derivations.
>
> **Weaknesses 1 & Question 1:** We agree that comparing against recent reasoning models is crucial. We evaluated **DeepSeek-R1** and **GPT-4o** on the C-STS test set using Few-shot prompting (see Table below).
>
> | Method                     | Training Paradigm | Spearman  |  Pearson  |
> | -------------------------- | :---------------: | :-------: | :-------: |
> | GPT-4                      |     Few-shot      |   43.60   |     -     |
> | DeepSeek-R1 (new)          |     Few-shot      |   42.85   |   43.36   |
> | GPT-4o (new)               |     Few-shot      |   44.23   |   44.07   |
> | PoLi-RL (Qwen3-0.6B) (new) |        RL         |   44.34   |   44.36   |
> | PoLi-RL (Qwen3-4B) (new)   |        RL         |   46.23   |   46.19   |
> | PoLi-RL (Qwen3-8B)         |        RL         | **48.18** | **48.27** |
>
> Our 8B model significantly outperforms **GPT-4o (+3.95)** and **DeepSeek-R1 (+5.33)**. Even our **0.6B model** slightly edges out these massive proprietary models. This suggests that while proprietary models possess strong general reasoning capabilities, they struggle to strictly align with the fine-grained quantization standards of C-STS (1-5 scale) in a few-shot setting. PoLi-RL bridges this gap by explicitly optimizing this alignment via RL, proving that a specialized, smaller model can surpass general-purpose giants on complex conditional ranking tasks.

---

> ### Author Response · Authors · 2025-11-21
>
> **Weakness 2 & Question 2:** We demonstrate that PoLi-RL is a versatile framework adaptable to diverse ranking paradigms beyond the specific C-STS setting. We address the generalizability concern from two complementary perspectives:
>
> - **A. Theoretical Adaptability to Local Ranking (Per-input List)**: Regarding the "per-input candidate set" scenario, we clarify that PSRR is mathematically **agnostic** to list construction. In **Local Ranking** tasks (e.g., MS MARCO), where the goal is to rank $K$ candidates for a single query $q$, PoLi-RL adapts seamlessly by redefining the **List Domain** from the *cross-sample batch* to the *per-input candidate set*. The mechanism remains identical: generate $G$ completions for each of the $K$ candidates and compute listwise rewards within that candidate slice. This indicates that PoLi-RL is technically compatible with "per-input candidate set" ranking without structural changes.
>
> - **B. Empirical Generalizability on Global Ranking (WMT-QE Task)**: To empirically prove robustness against C-STS-specific artifacts (e.g., 1-5 scale, adjacent pairs), we applied PoLi-RL to the **WMT-QE 2020** task [1]. This task also optimizes Global Spearman but differs significantly in setup:
>
>   - **Different Domain:** Multilingual Translation Quality.
>   - **Different Scale:** Labels are **0-100** (quality score averaged by annotators), breaking the "1-5" constraint.
>   - **Different Structure:** No adjacent pairs exist.
>
>   We removed the dataset-specific Pairwise and Binary rewards, relying solely on Stage I (Pointwise) and Stage II (Listwise PSRR) using the prompt setting from [2].
>
>   | Method             | Training Paradigm | Spearman  |  Pearson  |
>   | ------------------ | :---------------: | :-------: | :-------: |
>   | Qwen3-8B           |     Few-shot      |   45.03   |   44.18   |
>   | Qwen3-8B           |        SFT        |   50.90   | **51.09** |
>   | PoLi-RL (Stage I)  |        RL         |   51.72   |   50.58   |
>   | PoLi-RL (Stage II) |        RL         | **54.33** | **51.09** |
>
>   PoLi-RL achieves a **+3.43** Spearman gain over the SFT baselines. This empirically  validates that our PSRR mechanism is effective across different label scales and data organizations, independent of the dataset-specific reward components.
>
> [1] Unsupervised Quality Estimation for Neural Machine Translation. TACL 2020
>
> [2] TMU-HIT’s Submission for the WMT24 Quality Estimation Shared Task: Is GPT-4 a Good Evaluator for Machine Translation? Proceedings of the Ninth Conference on Machine Translation 2024

---

> > ### Author Response · Authors · 2025-11-21
> >
> > **Weakness 3 & Question 3:** We address your concerns by clarifying the theoretical grounds of our **rewards**, providing the rationale for **hyperparameters**, and presenting new component-wise **ablations**.
> >
> > - **A. Theoretical Grounds: Decomposing the Spearman Objective**. Regarding your question on why our reward aligns with the target metric, we ground this in the **decomposition** of the Spearman metric. Mathematically, maximizing Spearman’s $\rho$ is equivalent to minimizing the squared rank differences ($\sum d_i^2$), as defined by: $\rho = 1 - \frac{6 \sum d_i^2}{n(n^2-1)}$. Recognizing that directly optimizing this global objective yields  complex and coarse-grained signals that overwhelm the model, we decompose the intractable goal into three manageable levels:
> >
> >   - **Listwise (via PSRR):** We first break the global batch into parallel slices using PSRR. Within each slice, we optimize the rank errors ($\sum |d_i|$), serving as a dense proxy for the Spearman objective.
> >   - **Pairwise:** Since a correct permutation is functionally constructed from correct local orderings, we further decompose the list into adjacent pairs considering the  structure of C-STS dataset.
> >   - **Pointwise:** Finally, we decompose the pair into individual samples to anchor the absolute scores. Here, $R_{binary}$ prevents the model from hacking the metric by collapsing to the "safe" middle score (3), while $R_{pointwise}$ refines the scalar precision.
> >
> > - **B. Rationale for Hyperparameters**. We clarify that our settings are derived from data statistics (Normalization), not arbitrary choices.
> >
> >   - Normalization Constants (**Non**-hyperparameters):
> >
> >     - "$\text{Max}(Y)-\text{Min}(Y)=4$" (Eq. 5): Standard normalization to scale the pointwise reward to $[0, 1]$.
> >     - "$\text{Max Difference} = 3$" (Eq. 7): This is derived from the C-STS label range ($[1, 5]$). For any pair, the maximum possible ground-truth difference is $|5-1|=4$ , the minimum is $1$, so the max difference is $3$. We use this constant to normalize the pairwise reward to $[0, 1]$.
> >
> >   - Hyperparameters Selection: The table below summarizes our default settings. In this paper, we only tuned $\mu_1:\mu_2:\mu_3$ and slice size $N$ , other hyper parameters were originally set based on heuristics.
> >
> >     |         Hyper-Parameter         | Default Value |
> >     | :-----------------------------: | :-----------: |
> >     | $\lambda_1:\lambda_2:\lambda_3$ |   1:0.25:1    |
> >     |       $\mu_1:\mu_2:\mu_3$       |    1:1.5:1    |
> >     |           $R_{base}$            |      0.5      |
> >     |               $G$               |       8       |
> >     |               $N$               |      24       |
> >
> >     - Stage I Weights (1:0.25:1): We treat the Pointwise reward as the primary objective. The Binary reward serves as an **auxiliary** to prevent reward hacking.
> >     - Stage II Weights (1:1.5:1): We started with an equal weighting (1:1:1). Through tuning, we found that boosting the Pairwise weight to 1.5 yielded better performance.
> >     - $R_{base}$: This parameter balances the importance of **Ordinality** (getting the direction right) and **Cardinality** (getting the exact score difference right). We initialized it at $0.5$ to assign equal prior importance to both objectives.
> >     - Slice Size($N=24$):  We tuned $N$ ($32 \to 24$) on the validation set and found that a moderate batch size ($N=24$) is optimal.
> >     - Number of completions($G=8$): Represents the trade-off between Exploration Diversity and Computational Cost. We empirically found $G=8$ to be the optimal setting.

---

> ### Author Response · Authors · 2025-11-21
>
> - **C. Systematic Necessity Evidence**. We provide statistical tests and component-wise ablations.
>
>   - Significance Test: Since the official C-STS test set labels are withheld, we performed significance testing on the validation set. We employed Bootstrap Resampling to compare the Spearman score of PoLi-RL against the strongest SFT baseline. The test yielded a **p-value < 0.001** , confirming that the improvement of PoLi-RL over the baseline is statistically significant.
>
>   - Ablations (New Experiments): We conducted post-hoc ablations on previously fixed parameters.
>
>     - Number of generation ($G$):  Increasing $G$ from 4 to 8 improves Spearman but plateaus at $G=12$ (47.98). This confirms $G=8$ is the optimal trade-off between exploration diversity and computation cost.
>
>       | G            |       Method       | Spearman  |  Pearson  |
>       | ------------ | :----------------: | :-------: | :-------: |
>       | 4 (new)      | PoLi-RL (Stage I)  |   44.55   |   44.49   |
>       | 4 (new)      | PoLi-RL (Stage II) |   47.47   |   47.45   |
>       | 8 (original) | PoLi-RL (Stage I)  |   44.77   |   44.45   |
>       | 8 (original) | PoLi-RL (Stage II) | **48.18** | **48.27** |
>       | 12 (new)     | PoLi-RL (Stage I)  |   45.51   |   45.61   |
>       | 12 (new)     | PoLi-RL (Stage II) |   47.98   |   48.06   |
>
>     - Stage I  Reward Weights: Varying the auxiliary binary weight $\lambda_2$ from 0.1 to 1.0 yielded stable Spearman scores (44.67 ~ 44.94), proving robustness to this constraint.
>
>       | Method                       | $\lambda_1$(Pointwise) | $\lambda_2$(Binary) | $\lambda_3$(Format) | Spearman  |  Pearson  |
>       | ---------------------------- | :--------------------: | :-----------------: | :-----------------: | :-------: | :-------: |
>       | PoLi-RL (Stage I) (new)      |           1            |         0.1         |         0.1         | **44.94** | **44.93** |
>       | PoLi-RL (Stage I) (original) |           1            |        0.25         |         0.1         |   44.77   |   44.45   |
>       | PoLi-RL (Stage I) (new)      |           1            |         0.5         |         0.1         |   44.67   |   44.42   |
>       | PoLi-RL (Stage I) (new)      |           1            |          1          |         0.1         |   44.76   |   44.92   |
>
>     - $R_{base}$: The results confirm that 0.5 (equal importance to ordinality and cardinality) is optimal compared to removing it (0) or over-weighting it (0.75).
>
>       | Method             | $R_{base}$     | Spearman  |  Pearson  |
>       | ------------------ | -------------- | :-------: | :-------: |
>       | PoLi-RL (Stage II) | 0 (new)        |   47.90   |   47.59   |
>       | PoLi-RL (Stage II) | 0.25 (new)     |   47.78   |   47.81   |
>       | PoLi-RL (Stage II) | 0.5 (original) | **48.18** | **48.27** |
>       | PoLi-RL (Stage II) | 0.75 (new)     |   47.54   |   47.33   |

---

> > ### Author Response · Authors · 2025-11-27
> > **Update: Revised Manuscript**
> >
> > Dear Reviewer PEjZ,
> >
> > Thank you again for your constructive comments. We are writing to inform you that we have uploaded the **revised manuscript**. Based on your suggestions, we have formally integrated new experiments and **refined our claims**:
> >
> > - Hyperparameters settings and ablation study: Details in Appendix A.3.
> > - Reasoning Baselines: Added in Appdendix A.5.
> > - Performance on C-STS re-annotated data: Appendix A.7 demonstrates robustness under label noise.
> > - Generalization (WMT-QE Task): Appendix A.8 validates cross-domain adaptability.
> > - Conclusion Update: Integrating the positive findings from our generalization experiments (WMT-QE), we have revised the Conclusion to more precisely characterize the framework's contribution as "demonstrating the framework's potential adaptability to other ranking-based tasks".
> >
> > We hope these additional experiments and analyses satisfactorily address your concerns. We look forward to your thoughts on the updated paper.
> >
> > Best regards, Authors

---

### Official Review · Reviewer_fkXh · 2025-11-01

**Soundness:** 3
**Presentation:** 4
**Contribution:** 3
**Rating:** 8
**Confidence:** 4

**Summary:**

This paper introduces PoLi-RL, a novel Point-to-List reinforcement learning framework for Conditional Semantic Textual Similarity (C-STS). The work addresses the limitations of existing discriminative models by leveraging large language models (LLMs) and RL. To overcome the significant challenge of reward sparsity in this generative task, PoLi-RL reformulates the problem from a point-wise generation task into a list-wise ranking task. This approach effectively densifies the reward signal.

**Strengths:**

- The paper is built on a clear motivation and is well-written, with clean figures that effectively illustrate the proposed method.
- The method is a reasonable and well-designed approach that directly addresses the issue of sparse rewards, which is a key challenge for applying naive RL to this task.
- The experimental results are impressive. The paper provides comprehensive experiments and the corresponding analysis.

**Weaknesses:**

The authors did not conduct experiments on smaller generative models (e.g., qwen 0.5B) that would be more comparable in parameter count to the discriminative models used as baselines. It is unclear how much of the impressive performance gap is attributable to the method itself versus the model's scale.

**Questions:**

What is the base reward $R_{base}$ in equation (7)?

---

> ### Author Response · Authors · 2025-11-21
>
> We sincerely appreciate that you recognize the novelty of our PoLi-RL framework. We are particularly grateful for the insightful suggestion regarding smaller model scales, which has significantly strengthened our claims.
>
> **Weaknesses:** Following your valuable advice, we conducted a full set of new experiments using smaller **Qwen3-0.6B** and **Qwen3-4B** models.
>
> | Method               | Training Paradigm | Spearman  |  Pearson  |
> | -------------------- | :---------------: | :-------: | :-------: |
> | GPT-4                |     Few-shot      |   43.6    |     -     |
> | Qwen3-0.6B           |     Few-shot      |   25.25   |   25.19   |
> | Qwen3-0.6B           |        SFT        |   35.59   |   36.83   |
> | PoLi-RL (Qwen3-0.6B) |        RL         | **44.34** | **44.36** |
> |                      |                   |           |           |
> | Qwen3-4B             |     Few-shot      |   37.97   |   38.48   |
> | Qwen3-4B             |        SFT        |   38.41   |   39.45   |
> | PoLi-RL (Qwen3-4B)   |        RL         | **46.23** | **46.19** |
>
> The results provide compelling evidence for the efficiency of PoLi-RL. Our **0.6B model (44.34)** achieves a massive **+19.09** point improvement over its few-shot baseline and, remarkably, outperforms both the proprietary giant **GPT-4 (43.60)** and the previous Cross-Encoder SOTA **SEAVER (43.83)**. This confirms that the performance gap is primarily attributable to the PoLi-RL framework's ability to align reasoning with ranking metrics, rather than mere parameter scale. We observe a consistent scaling law where performance improves steadily from 0.6B to 8B, validating the method's stability across model sizes.
>
> ---
>
> **Questions:**  We apologize if the definition in Section 2.3 was not sufficiently explicit. $R_{base}$ is a hyperparameter (set to 0.5 in our experiments) that functions as a "guaranteed bonus" for predicting the **correct ranking direction**. Its purpose is to prioritize **Ordinality** (Direction) over **Cardinality** (Magnitude). We want the model to first learn who is higher, and then refine by how much.
>
> - If the model ranks a low-similarity pair higher than a high-similarity pair, the reward is **0**.
> - If the model correctly identifies the higher-scoring pair but the gap is inaccurate, it immediately gets **0.5** ($R_{base}$) plus a partial bonus based on the gap accuracy.
> - If both direction and magnitude are perfect, it gets **0.5** ($R_{base}$) + **0.5** (Max Bonus) = **1.0**.
>
> We will include this clarification in the revision.

---

> > ### Comment · Reviewer_fkXh · 2025-11-21
> > **Thank you for the rebuttal**
> >
> > Thank you for the rebuttal. The performance gap between SEAVER (355M, 43.83) and PoLi-RL qwen3-0.6b (44.34) is not significant now, which is reasonable, though. Since current auto-regressive LLMs are already scaled to multiple sizes, the proposed method remains valuable.
> >
> > I'd like to see how other reviewers respond to the rebuttal. And I keep my current score since it's already high.

---

> > > ### Author Response · Authors · 2025-11-27
> > > **Update: Revised Manuscript**
> > >
> > > Dear Reviewer fkXh,
> > >
> > > We sincerely thank you for your response and continued support. We have uploaded the revised paper, which incorporates the new experiments on model scaling and other ablation studies into the Appendix. Thank you again for your valuable time and championing our work.
> > >
> > > Best regards, Authors

---

### Author Response · Authors · 2025-11-30
**Summary of Rebuttal & Key Revisions: Scalability, Generalizability, and Methodological Rigority**

Dear Area Chair,

We thank the reviewers for their valuable feedback. During the rebuttal, we have added  new appendices (A.3–A.8) and extensive experiments to address concerns regarding scalability, generalizability, and methodology. We summarize the three fundamental updates below:

### 1. Scalability & Performance Attribution: Small Models Surpass Proprietary Giants (Addressing R1, R2, R3)

Reviewers questioned whether gains stemmed from the method or model scale, and requested comparisons with proprietary reasoning models.

- **Update:** We benchmarked PoLi-RL on smaller **Qwen3-0.6B/4B** backbones and compared them against **DeepSeek-R1** and **GPT-4o**. (New Appendix A.4, A.5)
- **Result:** As shown in the new **Tables 9 & 10**: Our 0.6B model (**44.34**) outperforms the previous SOTA SEAVER (**43.83**), and DeepSeek-R1 (**42.85**),  while also edging out GPT-4o (**44.23**).

- **Conclusion:** This empirically proves that the gains stem from the framework's effectiveness rather than parameter scale.

### 2. Generalizability & Data Robustness (Addressing R2, R3)

Concerns were raised regarding the method's generalizability beyond C-STS specific artifacts (e.g., 1-5 scale, adjacent pairs) and its robustness against label noise.

- **Update:** We extended PoLi-RL to the out-of-domain **WMT-QE 2020 task** (continuous 0-100 scale, no pairs), and evaluated on the newly released **Re-annotated** C-STS dataset (New Appendix A.7, A.8).

- **Result:**
  - **WMT-QE:** PoLi-RL achieves a **3.43** Spearman gain on WMT-QE using only the core PSRR mechanism (**Table 13**), validating that our mechanism is agnostic to data structures.
  - **C-STS Re-annotated Data:** PoLi-RL maintains a **3.99** lead over SFT on the re-annotated cleaner dataset (**Table 12**), confirming improvements are not an artifact of overfitting to noise.

### 3. Methodological Rigority: Theory, Hyperparameters & Ablations (Addressing R2, R3)

Reviewers requested experiment for the RL paradigm over differentiable ranking loss and the rationale for reward designs/hyperparameters.

- **RL vs. Strong Regression Baseline:** PoLi-RL (**48.18**) outperforms the strong Regression baseline with differentiable Pearson Correlation Coefficient (Pcc) Loss (**46.44**) (**Table 11**), validating that optimizing the reasoning process via RL is superior to black-box regression (New Appendix A.6).

- **Theoretical Grounding (Why these rewards?):** We clarified that our reward structure is a mathematical **decomposition of the Spearman objective**: Listwise (via PSRR) approximates global rank distribution, Pairwise ensures local ordinality, and Pointwise anchors scalar precision.
- **Extensive Ablations:** We detailed the hyperparameter setting and conducted extensive new ablations (New Appendix A.3).
  - **Generation Multiplicity ($G$):** Performance peaks at $G=8$, balancing exploration diversity with compute cost.
  - **Reward Weights in Stage I:** Varying the binary reward weight from 0.1 to 1.0 yielded robust performance.
  - **Base Reward ($R_{base}$):** Setting $R_{base}=0.5$ optimally balances ordinality (direction) and cardinality (magnitude).

**Closing Statement**  The rebuttal experiments reveal that PoLi-RL not only surpasses proprietary giants like GPT-4o on C-STS using the Qwen3-0.6B backbone, but also generalizes effectively to broader ranking tasks like WMT-QE. We believe the revised manuscript provides a compelling and rigorous contribution to the community.

Sincerely, The Authors

---

### Meta-Review · Area_Chair_BQG6 · 2025-12-19

**Summary:**

The paper introduces PoLi-RL, a two-stage point-to-list reinforcement learning framework for conditional Semantic Textual Similarity (C-STS) task with a cross-encoder backbone. The work aims to address the limitations of existing discriminative models often used in C-STS by introducing large language models (LLMs) and RL to C-STS. The framework includes two stages. Stage I teaches the model to score with pointwise distance, binary-consistency, and formatting rewards; Stage II adds pairwise and listwise ranking signals computed over parallel slices of completions (PSRR), yielding finer credit assignment aligned with rank-correlation objectives. With the two-stage design, the framework mitigate the challenge of reward sparsity when applying RL and LLMs to C-STS. Experiments on the standard C-STS benchmark (with an 8B-scale backbone) show consistent gains over few-shot prompting and supervised fine-tuning; ablations indicate that both the curriculum and the ranking-based rewards contribute.

Reviewers found the paper is of following strength

* It's interesting to introduce RL and LLMs to C-STS tasks, and the methods seem effective based on experiments provided in the work.
* Proposed method is reasonable and well-motivated. The method aims to address the challenge of sparse reward, which is the main bottleneck when applying RL to C-STS.
* Provided code and checkpoints for reproducibility.
* The paper is clear and well-written.

**Reviewer Concerns:**

At the beginning of the discussion, the reviewers had some questions including, lack of baselines (e.g., reasoning models, smaller generative models such as Qwen 0.5B), lack of evidence for generalizability, and relatively arbitrary design of rewards and hyperparameters.

**Reviewer Scores:**

Reviewers found the additional explanation from authors during rebuttal are helpful and explaining most of the questions. The additional insights, discussion, and experiment results greatly strengthen this work.

There is still one concern about the limitation of the scope. This work is heavily designed for C-STS, and it's not clear how well it can help general capabilities of LLMs. Compared to the main excitement about LLMs, its versatile capabilities in understanding and solving general tasks, this work seems too narrow and specialized. The reviewers are still worried that this work might not be interesting enough for the general ICLR audience and thus the reviewers are on the fence: reviewers recommend to accept this paper (by very slight margin) and ok for the authors to resubmit to a more suitable, NLP focused venue.

---

### Decision · Program_Chairs · 2026-01-26

Accept (Poster)